# Video Action Recognition Using Motion and Multi-View Excitation with Temporal Aggregation

**DOI:** 10.3390/e24111663

**Published:** 2022-11-15

**Authors:** Yuri Yudhaswana Joefrie, Masaki Aono

**Affiliations:** 1Department of Computer Science and Engineering, Toyohashi University of Technology, 1-1 Tenpaku-cho, Toyohashi 441-8580, Japan; 2Department of Information Technology, Universitas Tadulako (UNTAD), Palu 94118, Indonesia

**Keywords:** action recognition, multi-view, excitation, multi-layer neural network, temporal convolution, videos

## Abstract

Spatiotemporal and motion feature representations are the key to video action recognition. Typical previous approaches are to utilize 3D CNNs to cope with both spatial and temporal features, but they suffer from huge computations. Other approaches are to utilize (1+2)D CNNs to learn spatial and temporal features in an efficient way, but they neglect the importance of motion representations. To overcome problems with previous approaches, we propose a novel block which makes it possible to alleviate the aforementioned problems, since our block can capture spatial and temporal features more faithfully and efficiently learn motion features. This proposed block includes Motion Excitation (ME), Multi-view Excitation (MvE), and Densely Connected Temporal Aggregation (DCTA). The purpose of ME is to encode feature-level frame differences; MvE is designed to enrich spatiotemporal features with multiple view representations adaptively; and DCTA is to model long-range temporal dependencies. We inject the proposed building block, which we refer to as the META block (or simply “META”), into 2D ResNet-50. Through extensive experiments, we demonstrate that our proposed method architecture outperforms previous CNN-based methods in terms of “Val Top-1 %” measure with Something-Something v1 and Jester datasets, while the META yielded competitive results with the Moment-in-Time Mini dataset.

## 1. Introduction

Video action recognition is still challenging for researchers and practitioners as it involves both spatiotemporal and motion understanding. In the meantime, as an impact of the growth of technology, more people are involved in social media. They often record, upload and share videos to media platforms. As a result, an abundant number of videos are available to the public. This leads to more researchers engaging in the topic of video understanding. Action recognition, the first step of video understanding, becomes critical in practical applications such as suspicious behavior detection in camera surveillance and video recommendation systems.

An action in videos can be recognized based on scenes with less temporal information, while other actions need more temporal aspects to recognize. Such examples of actions with fewer temporal cues are *’rafting’* and *’haircut’*. We can judge the aforementioned actions only by seeing the scene. Contrarily, more temporal information is needed to judge an action involved in a video, e.g., *’zooming in with two fingers’* and *’picking something up’*. With this condition in mind, one must consider having both spatiotemporal and motion information flow in the network.

Current existing convolutional neural networks (CNNs) for action recognition can be categorized by the type of convolution kernel, i.e., three-dimensional (3D) and two-dimensional (2D) CNN. Several researchers utilized 3D CNNs to learn both spatial and temporal information simultaneously [1,2,3]. While this approach works very well for video action recognition tasks, the usage of a CNN-based 3D kernel certainly introduces more parameters compared with the 2D kernel type; hence, the computation cost increases. This will limit the implementation in real-time application. Additionally, 3D CNNs are typically laborious to optimize and prone to overfitting [3].

To overcome the real-time and optimization problem, recently, more researchers utilized 2D CNNs and equipped the network with temporal convolution for characterizing the temporal information [4,5,6]. TSN [7] employs a sparse temporal sampling strategy from the whole video to predict action in it, and this approach influenced several 2D CNN methods afterward. The approach by Lin et al. [6] utilized a shift-part strategy together with a sampling strategy to recognize an action very effectively. The shifting strategy is to shift part of the channels along the temporal axis in order to endow the network with movement learning. However, both approaches lack motion representations and spatiotemporal cues from different views. Meanwhile, since optical flow represents a motion as an input-level frame, many researchers take this modality into consideration. Two main drawbacks of optical flow are that it needs huge space storage and tremendous time for computation, which is not suitable for real-time application.

In different approaches from previous methods, several works factorized 3D cubes of a kernel into a (1+2)D kernel configuration [8,9,10] to lessen the heavy computation. Another interesting factorization strategy was introduced by the Deep HRI team [11] in the action recognition competition. They proposed their novel architecture, coined as multi-view CNN (MV-CNN), to act as a 3D convolution and showed a profound increase in the accuracy. Figure 1 illustrates their proposed idea and shows the comparison with two other convolution designs. Assuming *k* is the kernel with the size of 3: (a) the kernel convolves on temporal (*T*) and spatial (H×W) axes simultaneously; (b) temporal and spatial feature learnings are constructed serially; and (c) multi-view feature learnings occur independently, and the resulting feature maps are aggregated with weighted average $\alpha_{i}$ to produce multi-viewed feature maps. Nevertheless, these factorization strategies still neglect the importance of motion features, which are beneficial for action recognition tasks.

Based on the aforementioned observation, we propose a novel building block, **M**otion and Multi-View **E**xcitation and **T**emporal **A**ggregation (META). Specifically, META comprises three submodules: (1) Motion Excitation (ME), (2) Multi-view Excitation (MvE), and (3) Densely Connected Temporal Aggregation (DCTA). These three submodules are integrated into the 2D ResNet-50 base model, and it provides the network the ability to learn spatiotemporal and motion features altogether. The ME submodule addresses issues with optical flow problems by calculating feature-level content displacement on the fly during training or inferencing; thus, no space storage is needed. It also introduces an insignificant amount of FLOPs and time to calculate compared with optical flow. The MvE submodule, on the other hand, produces enhanced spatiotemporal representation of output features. This submodule adds a multi-view perspective to the original feature maps, and we design the MvE submodule to be complementary to ME, in that output from the MvE is directly added to the output from ME. For temporal feature learning, one-dimensional (1D) convolution is suitable for such a task. In our work, we insert densely connected 1D convolution layers inside a group of subconvolutions and arrange them in a hierarchical layout to model the temporal representation and long-range temporal dependencies. In a nutshell, the main contribution of our work is as follows:We design three submodules, including ME, MvE and DCTA, to learn enriched spatiotemporal and motion representations in a very efficient way and in an end-to-end manner.We propose META and insert it in 2D ResNet-50 with a few additional model parameters and low computation cost.We conduct extensive experiments on three popular benchmarking datasets, including Something-Something v1, Jester and Moments-in-Time Mini, and the results show the superiority of our approach.

## 2. Related Work

In this section, we discuss in more detail several related works for action recognition, including MV-CNN and TEA, which highly motivated this work. We also add a Transformer-based approach in this section because it has contributed to recent advances in computer vision, particularly for the task of action recognition.

With large-scale video datasets for action recognition available publicly, more 3D CNNs are introduced for action recognition tasks. Researchers successfully implemented them and outperformed state-of-the-art methods [3,9,12,13,14,15]. Three-dimensional CNNs are thought to be capable of learning both spatial and temporal aspects of a video, which are a very important aspect of action recognition. Among these researchers, Carreira et al. [13] used an ImageNet [16,17] pre-trained cube-shaped –N×N×N kernel to learn volumetric features. Another work by Hara et al. [15] used a 3D version of vanilla ResNet and proved the superiority of this architecture. Several attempts by [18,19,20] investigate the combination of a CNN and LSTM [21] or a densely connected LSTM. They claim that the LSTM layer can model the temporal relation of a series of features coming from either 2D CNNs or 3D CNNs. An alternative to LSTM for modeling temporal relationships is to use post hoc fusion [5] inside 2D CNNs. Later, a Temporal Shift Module (TSM) by Lin et al. [6] implemented a shifting operator on channel axes to learn temporal features without extra parameters and insignificant extra FLOPs. Afterwards, works proposed by [8,9,10] decomposed 3D convolution into 1D and 2D convolution to distill temporal and spatial features, respectively. This strategy was used to address the heavy computation and quadratic growth of model size when using 3D convolution. A mix of 2D and 3D CNNs in a unified architecture is also used to capture spatial and temporal features at the same time [22]. ECO [23] utilized this mixing strategy with top-heavy architecture. However, mixing the 2D CNN with the 3D CNN in one architecture will inevitably increase model parameters and require more time to optimize the parameters.

Meanwhile, partially inspired by the visual system of biological studies on the retinal ganglion cells in primates, Feichtenhofer et al. [24] advocated a two-stream architecture approach. Considering a video may contain more static as well as more dynamic objects, both streams have a different temporal rate, which makes their architecture unique compared with the existing two-stream architectures. Furthermore, learning from the three anatomical planes of the human body, i.e., sagittal, coronal and transverse, the DEEP HRI team [11] tried to simulate 3D convolution in their work. A video clip can be represented as a T×H×W volumetric data, with *T*, *H* and *W* denoting the number of frames, height and width, respectively. They reshape it to 2D data (i.e., T×H, T×W and H×W) separately and operate a shared 2D kernel to convolve over the reshaped 2D data. Figure 2 depicts images when they are seen in different views.

Apart from previous work, SENet [25] also attracts researchers to employ its squeeze-and-excitation method. This method uses a global context pooling mechanism to enhance the spatially informative channels and was verified to be effective in image understanding tasks. A recent work by Hao et al. [26] studied the insertion of channel context into the spatio-temporal attention learning block for element-wise feature refinement.

In addition to spatiotemporal representation, motion information is also the key difference between action classification and image classification tasks; thus, exploiting motion features is mandatory. A comprehensive study by Sevilla-Lara et al. [27] analyzed the importance of optical flow for action recognition. They analyzed optical flow itself, conducted several experiments using optical flow modality, and proved the contribution to the accuracy. Another several attempts utilized optical flow as an additional input to RGB [7,28]. A two-stream architecture is deployed to learn both optical flow and RGB data, and an average result is generated to predict an action. Feichtenhofer et al. [29] experimented with several fusion schemes so as to enhance spatiotemporal features. While this approach demonstrates excellent results compared with RGB data alone, this approach cannot be implemented in a real-world application, as extracting the pixel-wise optical frame with the TV-L1 method [30] requires heavy computation as well as much storage space. In this light, the RGB difference was introduced [7,31], which is more lightweight. A work by Jiang et al. [32] with their STM module firstly outlined the motion calculation for end-to-end learning in a 2D ConvNet and proved to be effective in capturing instantaneous motion representation as short-range temporal evolution. Furthermore, Li et al. [33] introduced a new block termed as TEA to explore the benefits of the attention mechanism added to the motion calculation previously mentioned. Later, this attentive motion features module was adopted by [34,35,36]. In addition, the authors of TEA suggested overcoming the limitation of long-range temporal representation by introducing multiple temporal aggregations in a hierarchical design. In this work, we propose a motion calculation and a hierarchical structure of local temporal convolutions, similar to the previous work. We explain more details of our work and highlight the difference from the previous work in the subsequent section.

As Vision Transformers brought recent breakthroughs in computer vision, specifically for action recognition tasks, many researchers have adopted them as their model [37,38,39,40] or combined them with 2D CNN [41]. For example, Arnab et al. [37] proposed several factorization variants to model spatial and temporal representation effectively inside a transformer encoder. TimeSformer [38] investigated several self-attention combinations on frame-level patches and suggested that separated attention for spatial and temporal representation applied within each block yielded the best video classification accuracy. Another work by Tian et al. [41] introduced a 2D ResNet with Transformer injected at the top layer before the linear layer to accurately aggregate extracted local cues from preceding blocks into a video representation. Although these current approaches seem promising, a Transformer-based network is not suited for real-world applications because it is highly computationally intensive [36].

## 3. Our Proposed Method

This section discusses the technical details of our work. Firstly, we present a method to extract motion representations to simulate the optical flow modality. Afterward, our novel Multi-view Excitation is introduced. Lastly, a simple stacking local temporal convolution with dense connection is also discussed here as a part of our improvement strategy. We also include a short discussion regarding comparing our work and TEA. Some notations written in this section are: *N*, *T*, *C*, *H* and *W*, indicating batch size, number of frames, number of channels, height and width, respectively.

### 3.1. Motion Excitation (ME)

Introduced firstly by STM [32], and later enhanced by TEA [33], the motion excitation submodule performs frame difference calculation in a unified framework for end-to-end learning. In principle, motion representation indicates content displacement of two adjacent feature maps, therefore called *feature-level*-based motion, rather than *pixel-level*-based motion, as in the concept of optical flow. Figure 3 illustrates the steps to measure approximate feature-level temporal differences.

The first step is to reduce the number of channels for efficiency with the ratio *r* = 16 by applying a 1×1 convolution layer $K_{red}$ to the initial input *X*, formulated in Equation (Equation 1). Then, we slice feature maps at the temporal axis, followed by element-wise subtraction for every adjacent output feature and obtain *M* at time step *t*. Before subtraction, a 3×3 transformation convolution layer $K_{transf}$ is applied to the output features $X^{\prime }$ at the time step (t+1). Next, we concatenate motion representations *M* at all time steps according to the temporal axis with 0 padded to the last segment. Concretely, given $X\in R^{NT\times C\times H\times W}$ are input features to the ME submodule, the above processes are expressed as follows: (1)$$\begin{array}{cc} X^{\prime } & =K_{red}*X,\;\;\;X^{\prime }\in R^{NT\times \frac{C}{r}\times H\times W} \end{array}$$(2)$$\begin{array}{cc} M_{t} & =K_{transf}*X_{t+1}^{\prime }-X_{t}^{\prime },\;\;1\geq t\geq T-1 \end{array}$$(3)$$\begin{array}{cc} X^{\prime } & =concat(M_{t},0),\;\;1\geq t\geq T-1, \end{array}$$
where $M_{t}\in R^{N\times \frac{C}{r}\times H\times W}$ and the last $X^{\prime }\in R^{NT\times \frac{C}{r}\times H\times W}$.

At this point, we have a new $X^{\prime }$ as approximate feature-level motion representations. Since we want to emphasize the informative features and suppress less useful ones alongside with [25], we squeeze the global information from each channel of the motion representations by utilizing the global spatial pooling layer. Then, another 1×1 2D convolution layer $K_{exp}$ performs channel expansion to restore the number of channels, and we obtain a new *X*, as in Equation (Equation 4). Lastly, attentive feature maps are obtained by feeding the new *X* to a sigmoid function δ, while final outputs $X_{ME}$ are produced from a multiplication between the initial inputs *X* and attentive feature maps, as defined by Equation (5).
(4)$$\begin{array}{cc} X & =K_{exp}*pool\left(X^{\prime }\right),X\in R^{NT\times C\times 1\times 1} \end{array}$$
(5)$$\begin{array}{cc} X_{ME} & =\delta \left(F\right)*X,\;\;\;X_{ME}\in R^{NT\times C\times H\times W} \end{array}$$

When subtracting the feature maps, we only calculate them one time: A collection of feature maps containing [2∼T] timestamps minus [1∼T−1].

### 3.2. Densely Connected Temporal Aggregation (DCTA)

Previously, learning temporal relationships in the task of action recognition was achieved by repeatedly stacking local temporal layers in deep networks. Unfortunately, it raises some problems. It is considered to be harmful to the features because the optimization message transmitted from distant frames has been weakened. To alleviate such a problem, we propose the Densely Connected Temporal Aggregation submodule. We follow the Res2Net design [42] to split feature maps in channel dimension into four subgroups of convolutions separately. Each subgroup consists of temporal and spatial convolutions configured serially, while one subgroup has temporal convolution only. In addition, output features from each subgroup flow to the next convolutional block and the neighboring subgroup through a residual connection, except for one subgroup without a residual-like connection (see DCTA submodule in Figure 4 for details). Thus, the last subgroup *aggregately* receives refined spatiotemporal features from former subgroups.

Regarding the temporal convolution, we arrange the layers in a stacked and *densely connected* fashion. Notably, its parameters are shared across subgroups. In this work, the number of temporal convolution layers for stacking is three, and these stacked layers are placed in three subgroups having a residual connection. More specifically, the first layer receives the encoded features from the summation of ME and MvE; the second layer receives input features from the first layer; and for the third layer, its input is formed from the summation of all the preceding layers’ output features. Formally,
(6)$$\left.\begin{array}{cc} X_{0}^{\prime } & =K_{temp}*X\\ X_{1}^{\prime } & =K_{temp}*X_{0}^{\prime }\\ X_{T}^{\prime } & =K_{temp}*\left(X_{0}^{\prime }+X_{1}^{\prime }\right) \end{array}\right\}$$
where $K_{temp}$, *X*, $X_{i}^{\prime }\in R^{NHW\times C\times T}$, $X_{T}^{\prime }\in R^{NHW\times C\times T}$ denote 1D convolution with a kernel size of 3, initial input features, output features from the *i*–th layer and the final result of the last temporal layer. We omit the necessary permutation and reshape *X* and $X_{T}^{\prime }$ for simplicity. After that, a 3×3 spatial convolution follows, as stated in the previous paragraph. For all subgroups in the DCTA submodules, the process can mathematically be expressed as:(7)$$\left.\begin{array}{cc} X_{0}^{\prime } & =K_{temp}*X\\ X_{1}^{\prime } & =K_{spa}*X_{T}^{\prime }\\ X_{2}^{\prime } & =K_{spa}*\left(X_{1}^{\prime }+X_{T}^{\prime }\right)\\ X_{3}^{\prime } & =K_{spa}*\left(X_{2}^{\prime }+X_{T}^{\prime }\right) \end{array}\right\}$$
where $X_{i}^{\prime }\in R^{NT\times \frac{C}{4}\times H\times W}$, $K_{spa}$ and $K_{temp}$ are output features of the *i*–th subgroup, spatial convolution and part-shift temporal convolution from [6], respectively. Lastly, we concatenate across channel dimensions to obtain the final output features $X^{\prime }$:(8)$$X^{\prime }=concat\left(\left[X_{i}^{\prime }\right]\right),\;\;\;\;i=\left[0,1,2,3\right]$$
where $X^{\prime }\in R^{NT\times C\times H\times W}$. Notice that in Figure 4, indices of subgroups from left to right are from 0 to 3, correspondingly.

### 3.3. Multi-View Excitation (MvE)

As illustrated in Figure 5, the MvE submodule has three branches to extract beneficial information from different views, similar to that in [11]. Given an input feature $X\in R^{NT\times C\times H\times W}$ for branch TH, we utilize a 1×1 2D convolution layer $K_{red}$ to reduce the channel number for efficiency with a ratio of *r* = 16, identical to Equation (Equation 1). Then, the tensor dimension is reshaped to comply with the desired dimension, i.e., $NT\times \frac{C}{r}\times H\times W\to NW\times \frac{C}{r}\times T\times H$. After that, a shared *channel-wise* convolution layer *K* is utilized to produce transformed feature maps $X_{TH}^{\prime }$. Formally,
(9)$$X_{TH}^{\prime }=K*X^{\prime },\;\;\;X_{TH}^{\prime }\in R^{NW\times \frac{C}{r}\times T\times H}$$

The last step is to reshape back the tensor dimension, i.e., $NW\times \frac{C}{r}\times T\times H\to NT\times \frac{C}{r}\times H\times W$. The rest of the branches are processed accordingly to produce $X_{TW}^{\prime }$ and $X_{HW}^{\prime }$. If we have obtained all the outputs from the other two branches, then the new $X^{\prime }$ is a convex combination of the $X_{i}^{\prime }$:(10)$$X^{\prime }=\sum_{i}\alpha_{i}*X_{i}^{\prime },\;\;\;i\in \left[TH,TW,HW\right]$$
where α is a weighted average with constraints of $\sum_{i}\alpha_{i}=1$ and each of the $\alpha_{i}\geq 0$. We argue that each branch will contribute differently to the performance of the model. The rest of operations are identical to Equations (Equation 4) and (5) to obtain attentive multi-view feature maps $X_{MvE}$.

The initial work of the multi-view design was proposed by Li et al. with their team DEEP HRI [11]. Different from their work, our work introduces the excitation algorithm to the MvE submodule so that it has a kind of attention mechanism.

### 3.4. Meta Block

For comparative purposes, we adopt 2D ResNet-50 as a backbone like other state-of-the-art methods [33,34,35]. As shown in Figure 6, each “conv2” in all residual blocks (conv2_x until conv5_x) is replaced by META. In total, we insert 16 blocks of META to endow the network with the ability to learn both spatiotemporal and motion representations efficiently. When feeding feature maps to the DCTA submodule, we sum all of the output features generated from ME, MvE and the former convolution block (denoted by $X_{ME}$, $X_{MvE}$ and *X*, respectively) to obtain $X^{\prime }$.
(11)$$X^{\prime }=X_{ME}+X_{MvE}+X,\;\;\;X^{\prime }\in R^{NT\times C\times H\times W}$$

### 3.5. Discussion with TEA

We want to highlight the differences between our work and TEA in this subsection. TEA adopted feature-level motion representation and enhanced it by excitation strategy with negligible extramodel parameters. Unlike TEA, which only considers *X* in parallel with the output of ME, we also added output features from the MvE submodule, as in Equation (Equation 11). Moreover, the network enjoys richer spatiotemporal and motion representation features since we re-calibrate the features by *both* ME and MvE submodules.

Regarding temporal aggregation inside the Res2Net module, TEA adopted it to enable their network to model the long-range spatiotemporal relationship by adding a local temporal convolution to each subgroup of convolution. However, in our work, we also added local temporal convolutions in each subgroup of convolution and arranged them in a *stacked up* and *densely connected* manner.

## 4. Experiment and Evaluation

In the following section, we explain our experiments in detail. Firstly, we describe the datasets we used and explain how we implement training and testing strategies, including hyperparameter settings. We also perform certain ablation experiments to investigate the contribution of each component of META. Later, we present the results and analysis along with the discussion.

### 4.1. Datasets

Our proposed method is evaluated on three large-scale action recognition benchmark datasets, i.e., Something-Something v1, Jester and Moments-in-Time Mini.

An action classification on the Something-Something v1 [44], a motion-centric type of dataset with 174 classes, requires temporal understanding to classify an action. This dataset is designed to emphasize the interaction between human and object, for example, “*Throwing something*” and “*Throwing something in the air and catching it*”. It contains 108,499 videos, with 86,017 in the training set and 11,522 in the validation set. Jester [45], which is also considered a temporal-related dataset, consists of 118,562 training videos, 14,787 validation videos and fewer categories than the Something-Something v1 dataset, i.e., 27. Example actions are “*Swiping up*” and “*Zooming out with two fingers*”. The Moments-in-Time Mini dataset [46] is a large-scale human-annotated collection of one hundred thousand short videos corresponding to dynamic events unfolding within three seconds; “*boxing*” and “*repairing*” are the two examples of categories. This dataset provides 100,000 videos for training and 10,000 for validation. It involves 200 action categories and offers a balanced number of videos in each category. While the previous datasets are more temporal-related, the Moments-in-Time Mini dataset can be considered both a temporal- and scene-related dataset. Frames have already been extracted from all videos in the Something-Something v1 and Jester datasets when they are made publicly available. However, in the Moments-in-Time Mini dataset, we must extract RGB frames from the videos at 30 frames per second at a resolution of 256 by 256. Figure 7 shows some images with their classes for the aforementioned datasets.

### 4.2. Implementation Details

#### 4.2.1. Training

We conduct all experiments on one Nvidia Quadro P6000 GPU card with PyTorch as the deep learning framework. We follow a sparse sampling strategy by TSN [7]. We extract *T* frames randomly from a number of evenly divided segments (in all our experiments, *T* = 8). Selected frames go through the network, and simple temporal pooling strategy is utilized to averagely predict an action for an entire video. Random scaling and cropping are applied as data augmentation. The size of the shorter side of the frame is cropped to 256 and resized to 224×224 to serve as the final frame size; hence, the final input shape is NT×3×224×224. Before the training started, we loaded our base model with weights trained on the popular ImageNet dataset [16,17]. As we adopt the Res2Net module for residing the DCTA submodule, we select the publicly available *res2net50 26w 4s* (available at: https://shanghuagao.oss-cn-beijing.aliyuncs.com/res2net/res2net50_26w_4s-06e79181.pth, accessed on 2 February 2022) pre-trained weights.

Regarding the hyperparameters for the Something-Something v1 and Moments-in-Time Mini datasets, the batch size, initial learning rate and dropout rate are set to 8, 0.0025 and 0.5, respectively. Moreover, the learning rates are decreased by factors of 10 at 30, 40 and 45 epochs and stops at 50 epochs. For the Jester dataset, the model is optimized for 30 epochs and the dropout is set to 0.5. Then, a learning rate is started at 0.0025 and reduced by factors of 10 at 10, 20 and 25 epochs. In addition, we unfreeze all instances of batch normalization layers during training. For the network optimizer, we select SGD with a momentum of 0.9 and a weight decay of $5\times 10^{-4}$. When setting the learning rate and weight decay for the classification layer on the three datasets above, we follow [33], i.e., 5× higher than other layers.

As a final thought, as suggested by [47], the learning rate must be matched with the batch size, i.e., the corresponding learning rate must be 2× higher when the batch size is scaled up by two. For example, if the learning rate changes from 2.5 $\times \;10^{-3}$ to 5$\times \;10^{-3}$, then batch size should increase from 8 to 16.

#### 4.2.2. Testing

We follow settings from [33] to adopt two methods as testing protocols: (1) efficient protocol, with frames × crops × clips is 8 × 1 × 1 and cropped 224 × 224 at central region as final frame size; and (2) accuracy protocol, with frames × crops × clips is 8 × 3 × 10, full resolution images (256 × 256 for final input size for frames) and averaged softmax scores for all clips for final prediction. When comparing with other recent works, we apply the accuracy protocol, as in Table 1 and Table 2. For the Moments-in-Time Mini dataset, as in Table 3, we apply the efficient protocol.

### 4.3. Results on Benchmarking Datasets

We report our experimental results and compare them with state-of-the-art methods. We list TSM to act as the baseline for Table 1 and Table 2. Since META is designed to function on CNN-based networks, we primarily compare our work with others whose networks are the same type as META to make relevant comparisons. Nevertheless, we still include recent Transformer-based networks in our comparison to demonstrate that META can achieve competitive accuracy while still being lightweight. We also include some successful predictions of META compared with other works on the three datasets we used.

#### 4.3.1. Something-Something V1

Something-Something v1 can be categorized as a temporal-related dataset; thus, ME and DCTA play important roles here. We divide Table 1 into four compartments; the upper part contains 3D CNNs, followed by 2D-based CNNs, Transformer networks and lastly, our model.

According to the table, our methods outperform the baseline for the 8 × 1 ×1 layout by sizable margins of 4.5% and 4.3% for top-1 and top-5 accuracy, respectively, while the FLOPs are only 1.08× higher. Our work is superior in that it significantly outperforms the baseline method’s 16 frames with a 2.9% accuracy improvement and low FLOPs, even when only eight frames and the one clip–one crop methodology are used. For the methods listed in the first compartment, we outperformed their work significantly. We considerably outscored I3D NL+GCN by only using eight frames, about 17× fewer FLOPs and fewer than half the parameters the I3D network used. A more competitive result is shown in the third compartment, where we outperform all current state-of-the-art methods in terms of top-1 accuracy. The nearest score to ours is TEA, where we obtain a substantially higher margin (52.1% vs. 51.7%), except top-5 accuracy is 0.3% lower (80.2% vs. 80.5%) when employing 10 clips. For comparison with SMNet [36], a more recent work, we noticeably outperform their work by big margins of 2.3% and 0.6% for top-1 and top-5 accuracy, respectively. This definitely demonstrates our superior submodules of MvE and DCTA combined with ME, considering SMNet also equipped their network with motion encoding.

When comparing our work with recent Transformer-based state-of-the-art methods, however, META is inferior to those methods presented in the middle part. Without considering FLOPs and the number of parameters, META is 3.3% less accurate than UniFormer-B in terms of top-1 accuracy and 1.3% lower than EAN RGB+LMC. According to [48], Transformer strength comes from its architecture, which was built to aggregate global information earlier due to self-attention. In addition to striking differences in architecture concept, we found that UniFormer used Kinetics as its pre-trained model, while we only pre-trained our model from ImageNet. Moreover, our model takes eight frames as the input image, while the Transformer-based models require more frames than us to serve as an input image.

Figure 8 shows a visual comparison of CNN-based techniques using a ball chart. We report the top-1 accuracy with respect to floating-point operations in gigabyte (GFLOPs). Accuracies are calculated using only center crop and single forward pass unless otherwise specified. The plot demonstrates how we consistently excel over comparable works while keeping FLOPs to a minimum level (only 1.08× as many as TSM). For our method and TEA, we find that total accuracy may be improved by a factor of ±1.05, at the expense of computational costs that increase to well over a thousand GFLOPs. The plot shows that overall, 2D CNNs may outperform 3D CNNs when the 2D-based network is provided with sufficient temporal feature learning.

#### 4.3.2. Jester

Likewise, Jester is classified as a temporal-related dataset. Our experiment result is provided in Table 2. Clearly, our work demonstrates superiority on the Jester dataset in terms of top-1 accuracy compared with the baseline (97.1% vs. 97.0%) and other state-of-the-art methods, except for ACTION-NET, where we obtained the same accuracy. Our interesting finding according to this table is: Both META and ACTION-NET, which are equipped with a motion representation module, achieved only a slightly higher accuracy than TSM (Δ0.1% of accuracy) without a motion representation module in it. Though, admittedly, we need further experiments to verify this, we think that motion encoding may have less meaning for this dataset. Moreover, results from TEA and MEST confirm our thoughts, as both methods proposed this module, and the performance is inferior compared with ours ({96.5%, 96.6%} vs. 97.1%).

Different from the previous dataset comparison, where our work has lesser predictive top-1 and top-5 accuracies than the methods utilizing Transformer, our work demonstrates very competitive results on the Jester dataset. META barely falls short of DirecFormer’s top-1 accuracy by 1.1% but surprisingly achieves a slightly higher top-5 accuracy (Δ0.2%). With the other two Transformer-based methods, we constantly outperform their works in top-1 and top-5 accuracies with significant gaps, proving the superiority of our work.

#### 4.3.3. Moments-In-Time Mini

Unlike the above datasets, this dataset possesses characteristics of temporal-related and scene-related datasets. The performance of our proposed work is still impressive, and Table 3 confirms this. We achieved the highest accuracy in terms of top-5 accuracy. While we obtain lower top-1 accuracy compared with IR-Kinetics400, we want to emphasize that IR-Kinetics400 utilized a Kinetics-400 [13] dataset as their pre-trained weights, whereas we only used ImageNet pre-trained weights. The closest accuracy to META is from I3D-DenseLSTM (Δ0.9% of top-1 acc.), where in their work, they utilized optical flow modality for encoding motion representations and LSTM to model long-range temporal representation, similar to META. Obviously, META is more efficient than I3D-DenseLSTM, as we estimate motion representations in a unified framework.

#### 4.3.4. Example of Successful Predictions

We illustrate some accurate predictions of META over other works in Figure 9. To obtain the probability score in (a), we re-train the model using the official code publicly available (https://github.com/Phoenix1327/tea-action-recognition) (accessed on 11 July 2022), whereas we only load the model with the official weights (https://github.com/V-Sense/ACTION-Net) (accessed on 11 July 2022) for (b). In all scenarios, we confidently achieve top-1 accuracy (indicated by a number in parenthesis) with substantial difference in probability score, whereas other works rank below ours. For instance, (a) informs that META exactly predict an action of “Lifting something with something on it” while TEA measures such action 8th out of the softmax outputs in descending order. This fact demonstrates our predominance over existing related works in three datasets.

#### 4.3.5. Learning Curve Analysis

During model training, we generate log statements of accuracies for each iteration and save them in a plain text file for further analysis. Figure 10 shows a training visualization in terms of top-1 and top-5 accuracies, with one crop–one clip for both training phase and inference. It is clear from the visualization that after 50 training epochs the model has not improved. Meanwhile, at epoch 30, the accuracy graph turned upward. This is due to our strategy of changing the learning rate at the epoch with our optimization method SGD.

### 4.4. Ablation Study

We perform some evaluations of our META comprehensively on the Something-Something v1 validation dataset and report the result in this subsection. All experiments utilize ImageNet pre-trained weights and are conducted using the efficient protocol. TSM serves as our baseline.

Impact of each moduleWe examine how each submodule affects the performance and present the findings in Table 4. It is clear that, in comparison with the baseline, each submodule continuously improves the performance of the 2D ResNet on video action recognition. The DCTA submodule makes the most contribution, improving top-1 accuracy by 2.4% while being computationally efficient with only a 1.7 G overhead gap and the least number of parameters, whereas the other two add 2.0 G of extra FLOPs.Location of METAWe examine the number of META implemented inside four convolution blocks toward accuracy. From Table 5, it is evident that better precision can be attained with more profound METAs placed in convolution blocks. Interestingly, META only requires installing one convolution block to dramatically increase the performance, with top-1 and top-5 accuracies exceeding the baseline by 2.6% and 2.9%, respectively.

## 5. Conclusions

This paper presents a novel building block to overcome the existing problems for the video action recognition task by designing three submodules to construct a META block and integrating it into each residual block of 2D ResNet-50. The proposed block includes excitation of motion and multi-view features followed by densely connected temporal aggregation. While retaining modest computations, our META achieves competitive results on three large-scale datasets compared with its 2D/3D CNN counterparts. Compared with recent Transformer-based networks, our work still achieves competitive results on the Jester dataset, while being inferior on the Something-Something v1 dataset. In the future, we would like to investigate another fusion approach, i.e., channel concatenation in the DCTA submodule, so that all layers are connected, and the current input is the concatenation of the preceding layers. This fusion will guarantee that new information is added to the collective knowledge.

## Figures and Tables

**Figure 1 entropy-24-01663-f001:**
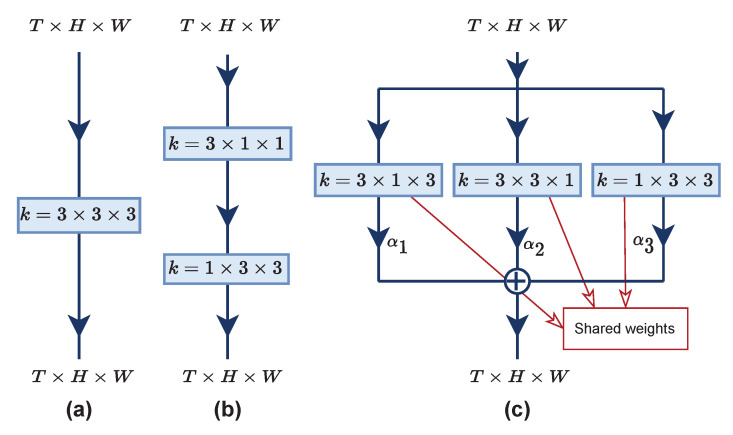
Comparison of three designs of spatiotemporal feature learning: (**a**) vanilla 3D convolution; (**b**) (1+2)D convolution; and (**c**) a multi-view design from [11].

**Figure 2 entropy-24-01663-f002:**
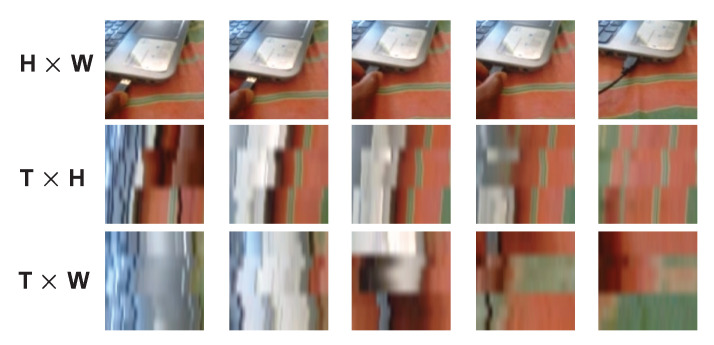
Image visualization with different views for action “Plugging something into something” class from the Something-Something v1 dataset.

**Figure 3 entropy-24-01663-f003:**
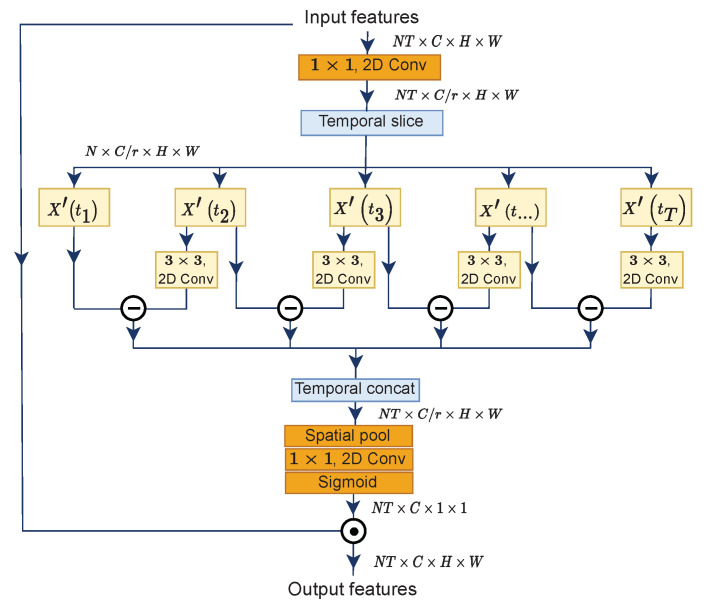
Two adjacent frames are subtracted to obtain motion representation. We firstly apply channel-wise 3 × 3 convolution on frames [t+1] before subtraction.

**Figure 4 entropy-24-01663-f004:**
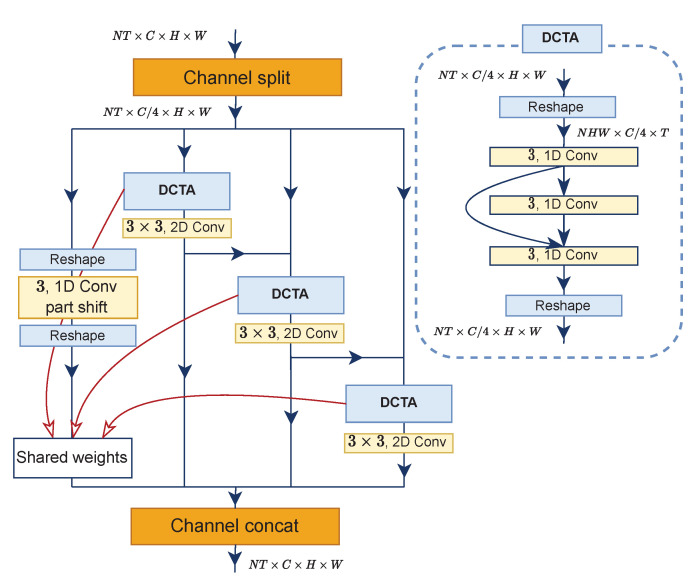
A detailed diagram showing the architecture of the DCTA submodule inside the Res2Net module.

**Figure 5 entropy-24-01663-f005:**
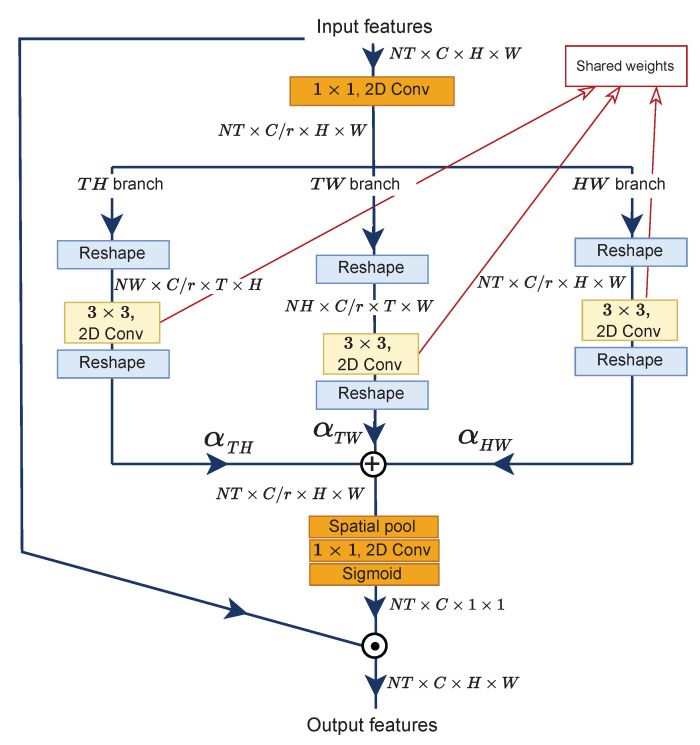
Detailed architecture of MvE submodule.

**Figure 6 entropy-24-01663-f006:**
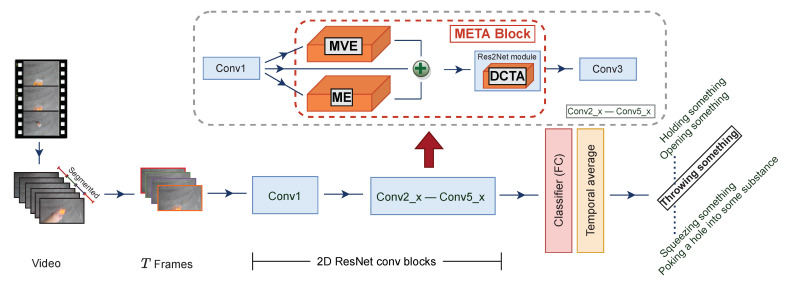
An overview of our proposed model implemented in 2D ResNet50 [43] architecture. We replace the original “conv2” with ME, MvE and DCTA inside every residual block to construct the META block. Inside the Res2Net [42] module, we insert a DCTA submodule. Details on data flow are given in Section 4.2.1.

**Figure 7 entropy-24-01663-f007:**
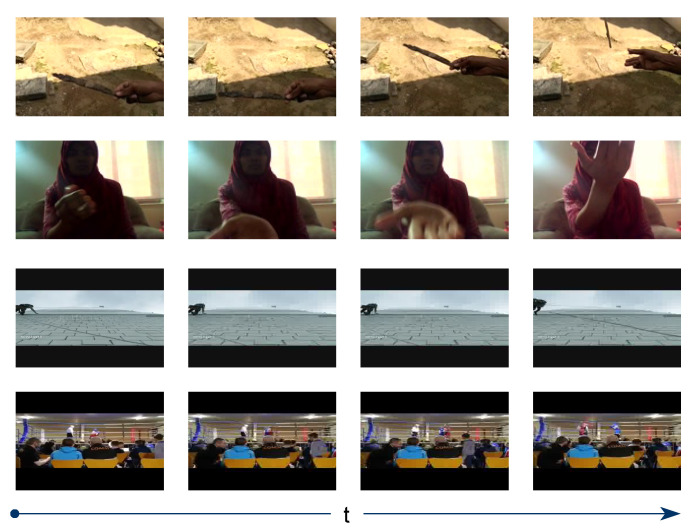
Examples of frames from (top-down) Something-Something v1 (“*Throwing something*”), Jester (“*Swiping up*”) and Moments-in-Time Mini (“*repairing*”, “*boxing*”) datasets.

**Figure 8 entropy-24-01663-f008:**
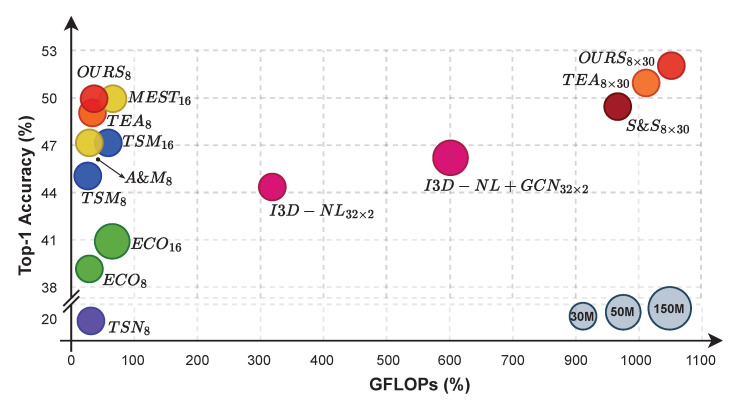
Ball chart reporting the top-1 accuracy *vs*. computational complexity (in GFLOPs). The size of each ball indicates model complexity. “A & M” corresponds to ACTION-NET [34] and MEST [35], while “S & S” denotes STM [32] and SMNet [36], respectively. We merge their icon since they share similar numbers of accuracy and GFLOPs.

**Figure 9 entropy-24-01663-f009:**
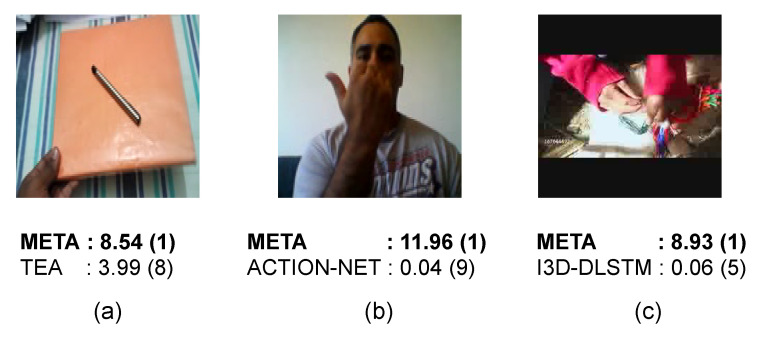
Examples of prediction results showing three frames and their probability scores (**a**–**c**): Something-Something v1 (“*Lifting something with something on it*”), Jester (“*Pulling hand in*”) and Moments-in-Time Mini (“*Tying*”) datasets.

**Figure 10 entropy-24-01663-f010:**
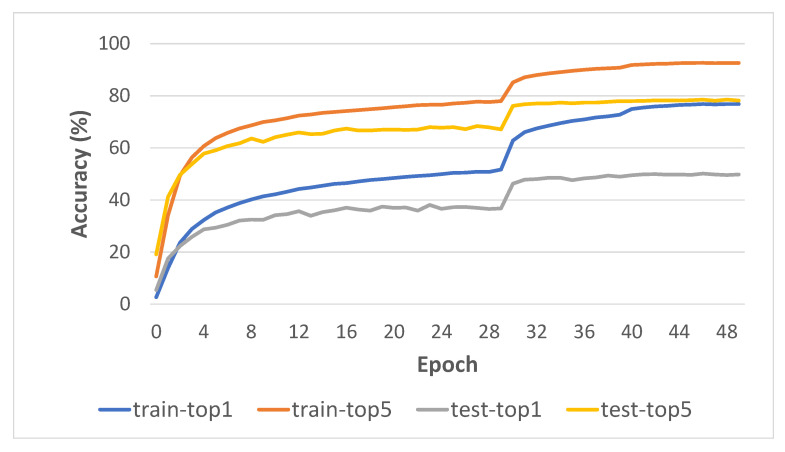
Training curve of our model on Something-Something v1 dataset.

**Table 1 entropy-24-01663-t001:** The comparison result of META against other state-of-the-art methods on the Something-Something v1 dataset. RN in the column backbone indicates ResNet. We list UniFormer methods with 16 input frames for relevant comparison. The highest accuracies for CNN-based networks are highlighted in bold.

Methods	Backbone	Pre-Train	Inputs	FLOPs	Param.	Top-1 (%)	Top-5 (%)
Three-Dimensional CNNs: ECO RGB from [6]	BNInception	Kinetics	8 × 1 × 1	32.0 G	47.5 M	39.6	–
ECO RGB from [6]	[1.2px]+ 3D RN-18		16 × 1 × 1	64.0 G	47.5 M	41.4	–
I3D NL RGB [14]	3D RN-50	ImgNet	32 × 1 × 2	168.0 G × 2	35.3 M	44.4	76.0
I3D NL+GCN RGB [14]		[1.2px]+ Kinetics		303.0 G × 2	62.2 M	46.1	76.8
Two-Dimensional CNNs: TSM RGB [6]	2D RN-50	ImgNet	8 × 1 × 1	33.0 G	24.3 M	45.6	74.2
TSM RGB			16 × 1 × 1	65.0 G	24.3 M	47.2	77.1
TSN from [6]			8 × 1 × 1	33.0 G	24.3 M	19.7	46.4
STM [32]			8 × 3 × 10	33.3 G × 30	24.0 M	49.2	79.3
STM			16 × 3 × 10	67.0 G × 30	24.0 M	50.7	80.4
TEA [33]			8 × 1 × 1	35.1 G ^2^	26.1 M ^2^	48.9	78.1
TEA			8 × 3 × 10	35.1 G × 30 ^2^	26.1 M ^2^	51.7	**80.5**
ACTION-NET [34]			8 × 1 × 1	34.7 G	28.1 M	47.2 ^3^	75.2 ^3^
MEST [35]			8 × 1 × 1	34.0 G	25.7 M	47.8	77.1
MEST			16 × 1 × 1	67.0 G	25.7 M	50.1	79.1
AIA TSM [26]			8 × 1 × 1	33.1 G	23.9 M	49.2	77.5
SMNet [36]			8 × 3 × 10	33.1 G × 30	23.9 M	49.8	79.6
Transformers:							
UniFormer-B [40]	Transformer	Kinetics	16 × 1 × 1	96.7 G	49.7 M	55.4	82.9
UniFormer-S [40]				41.8 G	21.3 M	53.8	81.9
EAN RGB+LMC [41]	Transformer + 2D RN-50	ImgNet	(8 × 5) × 1 × 1	37.0 G	36.0 M ^1^	53.4	81.1
Ours:							
META	2D RN-50	ImgNet	8 × 1 × 1	35.6 G	26.6 M	50.1	78.5
META			8 × 3 × 1	35.6 G × 3	26.6 M	51.0	79.3
META			8 × 3 × 10	35.6 G × 30	26.6 M	**52.1**	80.2

^1^ Not counting Latent Motion Code (LMC) module parameters. ^2^ Re-counted using official public code for digit precision. ^3^ Our implementation using official public code.

**Table 2 entropy-24-01663-t002:** Comparison with state-of-the-art methods on Jester validation set. These methods used 8 frames as model input. The highest accuracies for CNN-based networks are highlighted in bold.

Methods	FLOPs × Views	Top-1 (%)	Top-5 (%)
Two-Dimensioal ResNet-50:			
TSM [6]	33.0 G × 2	97.0	**99.9**
TSN from [6]	–	83.9	99.6
STM [32]	33.3 G × 30	96.6	**99.9**
TEA from [34]	–	96.5	99.8
ACTION-NET [34]	34.7 G × 30	**97.1**	99.8
MEST [35]	34.0 G × 2	96.6	**99.9**
Transformers:			
ViViT-L/16x2 320 [37] from [39]	–	81.7	93.8
TimeSFormer [38] from [39]	–	94.1	99.2
DirecFormer [39]	196.0 G × 3	98.2	99.6
META (Ours)	35.6 G × 30	**97.1**	99.8

**Table 3 entropy-24-01663-t003:** The comparison result of META against other CNNs on the Moments-in-Time Mini validation set. The highest accuracies are highlighted in bold.

Methods	Backbone	Top-1 (%)	Top-5 (%)
TRN from [49]	BNInception + InceptionV3	26.1	48.5
P3D from [49]	P3D ResNet	14.7	33.4
P3D-Kinetics from [50]	P3D ResNet	26.3	–
IR-Kinetics from [50]	Inception-ResNetV2	**30.3**	–
I3D-DenseLSTM [19]	I3D + ResNext	26.5	52.4
META (Ours)	2D ResNet-50	27.4	**53.2**

**Table 4 entropy-24-01663-t004:** The comparison result of an individual component against the baseline, including FLOPs and the number of parameters.

Methods	FLOPs	Param.	Top-1 (%)	Top-5 (%)
TSM [6]	33.0 G	23.7 M	45.6	74.2
ME	35.0 G	26.1 M	47.9	77.8
MvE	35.0 G	26.1 M	46.3	76.9
DCTA	34.7 G	25.7 M	48.0	77.0
META	35.6 G	26.6 M	50.1	78.5

**Table 5 entropy-24-01663-t005:** Examination of the quantity of location METAs inserted into 2D ResNet-50 residual convolutional blocks.

Location	Top-1 (%)	Top-5 (%)	Δ Top-1 (%)	Δ Top-5 (%)
TSM [6]	45.6	74.2	–	–
conv{2}_x	48.2	77.1	+2.6	+2.9
conv{2,3}_x	49.5	78.1	+3.9	+3.9
conv{2,3,4}_x	49.9	78.2	+4.3	+4.0
META	50.1	78.5	+4.5	+4.3

## Data Availability

Somthing-Something v1: Not applicable; Jester is available at https://developer.qualcomm.com/software/ai-datasets/jester (accessed on 2 February 2022); Moments-in-Time Mini: Not applicable.

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
