# Peer review of "Video Action Recognition Using Motion and Multi-View Excitation with Temporal Aggregation"

_entropy, 2022, doi:10.3390/e24111663_

Round 1
Reviewer 1 Report
This paper proposes a novel approach to video action recognition. They use a block, including motion excitation, multi-view excitation, and densely connected temporal aggregation, to capture spatial and temporal features faithfully and efficiently learn motion features. The proposed block was injected into 2D ResNet-50. The experimental results reveal that the proposed approach outperformed previous CNN-based methods in three different datasets. For this paper, I have some comments.
(1) This paper is well-structured and easy to read. From the abstract and introduction, the reader can quickly realize the focus and contribution of this paper.
(2) In the related work, all the cited references are related to the topics.
(3) To overcome the problems of computation time and lack of motion representation incurred by previous approaches, they use a novel block, including ME, MvE, and DCTA, to capture spatial and temporal features faithfully and efficiently learn motion features. The proposed method looks reasonable.
(4) I suggest that the explanation of Figures 1,3,7,8 should shift to the context instead of the caption.
(5) I suggest that the author should discuss the cost of adding this novel block.
Reviewer 2 Report
I think the authors have submitted a well-written manuscript which is illustrated with figures which help the understanding of the manuscript. I have rather minor comments to the manuscript.
a) In Subsection 4.1, the authors should summarize the main characteristics of the applied benchmark databases, i.e. number of videos, number of classes, resolution, fps, format, etc. This way the reader could get a better understanding of the results and databases.
b) In the Implementation details section, the authors should write also about the applied programming languages and libraries.
c) Since deep learning involves a lot of experiments, the publication of training curves would be nice.
d) The results of the comparison to the state-of-the-art should also be summarized visually (bar plot, etc.) besides Tables 1-3.
